# Modeling and Experiment for the Diffusion Coefficient of Subcritical Carbon Dioxide in Poly(methyl methacrylate) to Predict Gas Sorption and Desorption

**DOI:** 10.3390/polym14030596

**Published:** 2022-02-01

**Authors:** Jaehoo Kim, Kwan Hoon Kim, Youngjae Ryu, Sung Woon Cha

**Affiliations:** Department of Mechanical Engineering, Yonsei University, 50 Yonsei-ro, Seodaemun-gu, Seoul 03722, Korea; murja@yonsei.ac.kr (J.K.); kimkevin99@yonsei.ac.kr (K.H.K.); yjryu1027@yonsei.ac.kr (Y.R.)

**Keywords:** poly(methyl methacrylate), diffusion coefficient, gas sorption, solubility, solid-state batch foaming, gas desorption, curve fitting

## Abstract

Several researchers have investigated the phenomenon of polymer–gas mixtures, and a few have proposed diffusion coefficient values instead of a diffusion coefficient model. There is a paucity of studies focused on the continuous change in the diffusion coefficient corresponding to the overall pressure and temperature range of the mixture. In this study, the gas sorption and desorption experiments of poly(methyl methacrylate) (PMMA) were conducted via solid-state batch foaming, and the weight change was measured using the gravimetric method with a magnetic balance. The control parameters were temperature, which ranged from 290 to 370 K, and pressure, which ranged from 2 to 5 MPa in the subcritical regime. Based on the experimental data, the diffusion coefficient of the PMMA was calculated using Fick’s law. After calculating the diffusion coefficient in the range of the experiment, the diffusion coefficient model was employed using the least-squares method. Subsequently, the model was validated by comparing the obtained results with those in the literature, and the overall trend was found to be consistent. Therefore, it was confirmed that there were slight differences between the diffusion coefficient obtained using only Fick’s equation and the value using by a different method.

## 1. Introduction

Research on polymer–gas mixtures has traditionally been considered important. This research is related to the production of drug delivery devices, shoe soles, gas detection sensors, gas separation membranes, and foamed plastics [1]. Since the microcellular foaming process (MCP), which comprises the use of a supercritical fluid as a blowing agent, was developed at Massachusetts Institute of Technology in the early 1980s [2], the interest in this research subject has further increased in industries that use supercritical fluids [3]. The MCP is a technology that involves the creation of fine voids having a density of >10^9^ cells/cm^3^ within 10 μm inside a polymer by using a polymer–gas mixture and thermodynamic instability and has been used in a variety of industries, such as automobiles, sports, and packaging, as it has advantageous properties of specific strength [4], thermal insulation, reflectance [5], acoustic absorption, electromagnetic interference shielding, and lightweight ordinary plastic foaming. 

Typical manufacturing techniques for cellular foaming polymers include solid-state batch foaming, extrusion microcellular foaming, and injection mold microcellular foaming. The solid-state batch foaming method is mainly used at the laboratory scale because it is less productive than other methods, but it allows the easy control of variables such as temperature and pressure. Figure 1 illustrates the steps involved in the solid-state batch foaming process. Generally, cellular-foamed polymers are characterized by the cell size, cell density, and foaming ratio. These characteristics are dependent on each other [6] and are also closely related to the solubility of the polymer. The solubility is affected by the foaming temperature, foaming pressure, and saturation time. In general, as the temperature increases, the polymer solubility decreases, and as the pressure increases, the solubility increases. This is to Henry’s law. When the gas can no longer melt in the polymer after a sufficient amount of time, it is termed as fully saturated, and the ratio of the amount of molten gas to the polymer is defined as the solubility. The solubility depends on the temperature, pressure, and type of polymer. Furthermore, even the same polymer has different solubilities based on the properties of the polymer such as its chain length and molecular weight. Therefore, researchers examining plastic foaming initially measure the solubility and then conduct experiments. Many studies have been conducted on the solubility of various polymers including polycarbonate [7,8], polyethylene [9,10], polypropylene [11,12], polyurethane [13,14], and polystyrene [15,16], and such studies are still being conducted.

Poly(methyl methacrylate) (PMMA) is widely used not only in the industry but also in research owing to its high transparency [17,18,19], weatherability [20,21] and scratch resistance [22]. Moreover, given the carbonyl functional group of PMMA [23], the solubility of carbon is higher than that of other polymers. Hence, research on polymer–gas mixtures has been attracting significant attention. Many researchers have studied gas solubility using diffusion coefficients because diffusion is considered as a major mechanism in realizing a polymer–gas mixture. Edwards et al., reported the solubility of carbon dioxide in the PMMA using a chromatographic method [24]. Li et al., calculated the diffusion coefficient using the Arrhenius equation [25]. Ushiki et al., fitted solubility data for the diffusion coefficient using the Sanchenz–Lancomb equation of state [26]. 

Gas desorption is an important factor of consideration in plastic foaming. In the batch process, gas desorption refers to the amount of gas emitted over the time it takes for the polymer–gas mixture to be placed in high-temperature glycerin. This implies that the time between the step for creating the polymer–gas mixture and the cell nucleation step is termed as the gas desorption time. The dissolved gas in the polymer is released without causing thermodynamic instability because pressure decreases the solubility of the gas in the polymer. The desorption of gas in a polymer is not a reversible process of the sorption phenomenon [27]. Therefore, if gas desorption is not considered in plastic foaming, there will be a difference between the actual foaming ratios of the polymers. Therefore, the first step in accurately controlling the foaming characteristics involves controlling the gas sorption and desorption. However, while many researchers have studied gas solubility in polymers, there is a paucity of research on gas desorption, which proceeds via similar mechanisms.

In this study, we first conducted experiments on the solubility and gas desorption of the PMMA as a batch foaming process. Based on the obtained experimental data, a diffusion coefficient model was proposed using the least-squares method. Using this model, we predicted the solubility and gas desorption of the PMMA at various temperatures and pressures and compared them with the solubility data reported in other studies.

## 2. Materials and Methods

### 2.1. Materials

#### 2.1.1. Polymer

The material used in this study was PMMA (S-POLYTEC, Jincheon-gun, Chungcheongbuk-do, Korea). This material was a representative amorphous, glassy, thermoset polymer. The specimen was fabricated using the extrusion method. The dimensions of the specimen were 25 × 25 mm^2^, and to confirm the effect of the thickness, the thickness of the specimen was prepared as 1, 2, and 3 mm. The density of the specimen was 1.189 g/cm^3^, and the glass transition temperature was 393 K.

#### 2.1.2. Blowing Agent

In general, nitrogen and carbon dioxide are used as blowing agents for the polymers. The solubility of the majority of polymers is higher in carbon dioxide than in nitrogen under the same conditions. Therefore, in our study, carbon dioxide (Samhung GasTech, Seoul, Korea) was selected as the physical blowing agent for checking the solubility under various conditions.

### 2.2. Solid-State Batch Foaming Process

As the objective was to investigate the gas solubility and gas desorption, only the first stage of the solid-state batch foaming process was performed. That is, the steps for the cell nucleation and stabilization were omitted in this study. The experimental equipment we used is presented in Figure 2. The inner radius and height of the pressure vessel are 26 and 200 mm, respectively. The electric band heater is attached to the vessel to control the temperature during the gas sorption and desorption. The solubility and gas desorption were measured by weight using a gravimetrical method and a scale with an accuracy of 0.01 g (AR2130, OHAUS Corp, Parsippany, NJ, USA).

### 2.3. Diffusion Coefficient

The gas solubility and gas desorption were assumed to be caused only by the diffusion. Therefore, it is important to calculate the diffusion coefficient accurately for the prediction of the gas solubility and gas desorption. Diffusion is a mass transport mechanism by which particles or molecules move from a high-concentration region to a low-concentration region through arbitrary molecular motion. Fick proposed two laws for the diffusion process in 1855, the second of which is commonly referred to as the diffusion equation, which is expressed as shown in Equation (1) [28].
(1)$$\frac{\partial C}{\partial t}=D\frac{\partial^{2}C}{\partial x^{2}}$$
where C is the concentration of molecules, x is the position, and D is the diffusion coefficient. The following three assumptions were considered to analyze the diffusion coefficients of ${\mathrm{CO}}_{2}$ in PMMA: (i) the diffusion process follows Fick’s law; (ii) the diffusion process is performed only in the one-dimensional direction, which is the thickness direction of the PMMA sample. Furthermore, the diffusion from the side of the PMMA can be ignored and is assumed to occur only from the upper side of the PMMA; (iii) the change in PMMA thickness caused by the diffusion process is negligible. Equation (2) presents an analytical solution of the diffusion equation, which is presented in Equation (1), and can be obtained using the above assumptions and the Laplace transform [29].
(2)$$\frac{M_{t}-M_{0}}{M_{\infty }-M_{0}}=1-\frac{8}{\pi^{2}}\overset{\infty }{\underset{n=0}{\sum }}\frac{1}{{\left(2n+1\right)}^{2}}\exp\left(-\frac{{\left(2n+1\right)}^{2}\pi^{2}\mathrm{Dt}}{4L^{2}}\right)\;$$
where $M_{t}$, $M_{0}$, and $M_{\infty }$ are the amounts of dissolved $CO_{2}$ in the PMMA at the diffusion time t, initial time, and equilibrium, respectively. L denotes the thickness of the PMMA. It is unreasonable in terms of computational resources to consider all the infinity terms. Therefore, it is important to determine the number of terms that should be considered, and Li et al., calculated the above equation while considering one term [19]. To check the correlation between the number of terms and the error, the difference between the values of the right and left terms in Equation (2) at the initial value was analyzed according to the number of terms. When t = 0, that is, in the initial condition where the diffusion process does not occur, the right term becomes zero. Therefore, the value of the right term can thus be considered as an error, as shown in Figure 3. If the number of terms is greater than 20, the error is less than 1%. Therefore, in this paper, the diffusion coefficient was calculated by using up to 20 terms in the calculation.

## 3. Results and Discussion

### 3.1. Experiments for Gas Sorption

The main factors in the sorption experiment were the pressure and temperature. Because the target of this study is in the subcritical regime, the saturation temperature was varied from 290 to 370 K at intervals of 10 K, and the saturation pressure was varied from 2 to 5 MPa at intervals of 1 MPa. The saturation times were 3, 6, 12, 24, and 48 h, and for some cases, to understand the full saturation, the experiment was conducted for 120 h (5 d). The pressure gradient for depressurization was maintained at 1 MPa/s in all the experiments. Moreover, the time interval between the removal of the specimen from the pressure vessel and its weight measurement on an electronic scale was fixed at 2 min to minimize the desorption time.

#### 3.1.1. Effect of Specimen Thickness

The solubility tendency of the specimens with respect to the thickness over the saturation time was confirmed and is illustrated in Figure 4. Table 1 lists the diffusion coefficient for each thickness calculated using Equation (2) and the least-squares method. The experiments and calculations indicated that as the thickness of the specimen increases, the time to complete saturation increases. However, it did not affect the amount of CO_2_ that could dissolve per polymer volume, that is, the solubility. Therefore, a specimen thickness of 1 mm was selected for the subsequent experiment.

#### 3.1.2. Sorption Data

The experiment was conducted with a 1 mm specimen, the pressure range was 2–5 MPa, the temperature range was 290–370 K, and a total of 36 cases were subjected to the sorption experiment. Each case was conducted with seven specimens, and the average of the results, excluding the maximum and minimum values, was obtained. Figure 5a presents the sorption curves for various temperatures at 5 MPa. Figure 5b illustrates the sorption curves for various pressures at 290 K. The solubility data of the PMMA in a target temperature and pressure range are summarized in Table 2. The amount of CO_2_ that can be dissolved in the PMMA decreases as the temperature increases; the solubility of CO_2_ in PMMA follows Henry’s law. However, in the case of the full saturation time, it does not appear to be a linear relationship. This is because the higher the temperature, the lower the solubility, but the faster the diffusion rate. Consequently, there is a trade-off between the amount of CO_2_ that can be dissolved and the rate of dissolution in the PMMA.

### 3.2. Experiments for Gas Desorption

The atmospheric pressure of ${\mathrm{CO}}_{2}$ is considered as approximately zero for the process of gas desorption, and therefore, this process is only a function of the desorption temperature. It can be analyzed using Equation (2) because the only difference is the direction of the blowing agent, and it functions according to the same theoretical principle. Because $M_{\infty }$ is nearly zero and $M_{0}$ is the amount of gas sorption, Equation (2) can be expressed as Equation (3):(3)$$\frac{M_{t}}{M_{0}}=\frac{8}{\pi^{2}}\overset{\infty }{\underset{n\;=0}{\sum }}\frac{1}{{\left(2n\;+1\right)}^{2}}\exp\left(-\frac{{\left(2n\;+1\right)}^{2}\pi^{2}\mathrm{Dt}}{4L^{2}}\right)\;$$

The desorption experiment was conducted in a vessel with electric band heater, followed by the sorption experiment. The amount of time taken from the depressurization of the sorption experiment to placing the specimen in the temperature-controlled vessel was fixed at 2.5 min. The specimen was checked every 1 h for the first 12 h, and thereafter, it was checked every 6 h. It is known that it takes approximately 6 months for the gas in the polymer to be completely removed. Therefore, we extended the duration to 48 h, when 5–10% of the gas remained in the polymer to conduct a reasonable experiment. The desorption curves as a function of temperature are shown in Figure 6. At the same temperature, there was no difference in the diffusion coefficient, irrespective of whether the PMMA was fully saturated.

### 3.3. Diffusion Coefficients

#### 3.3.1. Swelling

During the process of generating the polymer–gas mixture, swelling occurs and the volume of the polymer increases. Because the dimension that has the greatest effect on the solubility is the thickness, the change in thickness depending on the solubility was investigated. The thickness of the specimen was measured using a digital microscope (Dino-Lite Edge, Model No. AM4515T8), and the change in the thickness relative to the solubility is presented in Figure 7. As expected, the higher the solubility is, the greater the thickness. This is due to the plasticization effect that the gas molecules dissolved in the polymer increases the distribution of the free volume size [30]. Using the experimental conditions with the highest solubility (5 MPa and 290 K, that is, 19.6% solubility), there was a change in the thickness of only 30 ± 1.5 μm, which correlates to a 3% change in the total specimen thickness. When the thickness changes by 1%, 2%, and 3% under the same diffusion coefficient conditions, the change in the sorption curve is as shown in Figure 8. The maximum relative error is 2.89%, but the relative error in the fully saturated regime, which corresponds to more than 12 h of saturation time, is approximately 0.5%. Therefore, the swelling is not considered when calculating and modeling the diffusion coefficient, which agrees with the previous assumption in Section 2.3.

#### 3.3.2. Modeling

The diffusion coefficient of the PMMA-CO_2_ for the sorption process, calculated using the data obtained through the experiment and Equation (2), is presented in Figure 9. Figure 10 illustrates the diffusion coefficient for the desorption process calculated using Equation (3). The desorption process is only a function of temperature, and curve fitting for one variable was performed. In contrast, because the sorption process is a function of both the temperature and pressure, it was fitted to the field rather than the curve.
(4)$$\begin{array}{ll} D_{\mathrm{sorption}}=-61.31 & +0.382T+27.47P-0.0005811T^{2}-0.1758\mathrm{PT}+0.5241P^{2}\\ & +0.0002685T^{2}P-0.0004917{\mathrm{TP}}^{2}-0.02809P^{3} \end{array}$$
(5)$$D_{\mathrm{desorption}}=0.001596T^{2}-0.9723T\;+150.8$$

Equations (4) and (5) show the diffusion coefficients according to the sorption and desorption processes, respectively. For the sorption curves, the sum of square errors (SSE) has a value of 1.354, and R-square is 0.8472. The SSE is a preliminary statistical calculation for the regression method and means a value that the proposed model cannot predict. The diffusion coefficient of the sorption process appears to be closely related to the effective glass transition temperature. Compared to other conditions, the diffusion coefficient at 5 MPa is very high. It depends on the grade of the PMMA, but at a pressure of ≥5 MPa, it generally comprises a rubbery state. In this rubbery state, the movement of molecular segmentation increases, and the free volume increases; thus, the gas tends to dissolve in the PMMA more easily, resulting in a higher diffusion coefficient. In contrast, the diffusion coefficient value at 2 MPa is very low as compared to other conditions. This seems to be because the state of the PMMA is always located in the left region of the effective glass transition temperature curve under the temperature conditions of this experiment, i.e., it is in the glassy state [31]. Overall, there is a large change in the tendency of the diffusion coefficient between 290 and 300 K, which appears to be consistent with the tendency of transition from the rubbery state to the glassy state. This is due to the retrograde behavior that appears in the PMMA-CO_2_ mixture which has two glass transition temperatures in the same pressure.

The SSE has a value of 0.8201 and R^2^ is 0.983 in the desorption curve modeling. Because the state transition does not occur at 0 MPa, there is no significant change in the trend. Therefore, it is apparent that the diffusion coefficient tends to increase as the temperature increases.

#### 3.3.3. Validation

Validating the results of research comprising a numerical analysis or modeling is crucial. Data from other studies conducted under the same conditions as those in this study were used as verification targets, which is presented in Figure 11.

The values differ according to the grade of the PMMA [35]. It was confirmed that the overall tendency was similar, although the value itself was different when compared with the obtained values presented in previous literature.

## 4. Conclusions

In this study, sorption and desorption experiments were conducted between PMMA and CO_2_ in the subcritical regime. Using these data, models for the diffusion coefficients of sorption and desorption were proposed. The diffusion coefficients were calculated based on Fick’s diffusion law, and the hypotheses used in the process were verified through experiments for observing the effects of swelling and thickness. It was found that the solubility and swelling were proportional, and when the solubility was 19.6%, the change in thickness was only 3%. Moreover, it was confirmed that the dimensions of the specimen, that is, thickness, width, and length, were independent of the diffusion coefficient.

The models were validated based on the results of previous studies, and though there was a difference in the exact values of the solubility and diffusion coefficients, the overall trend was consistent. Particularly in the case of the diffusion coefficients, there was little difference in the calculated values of this study and those of Guo [34], who conducted the experiment with a specimen of 1.48 mm. 

In general, it is known that the diffusion of the polymer cannot be explained using only Fick’s equation for a wide range of temperature and pressure conditions [36]. However, in this study, it was confirmed that relatively accurate values can be obtained in the subcritical regime even if only Fick’s equation is used. In addition, this method of calculating the diffusion coefficient may be applied not only to PMMA but also to other polymers. It is believed that the experimental stage for the solubility—an essential stage before conducting an experiment on a polymer–gas mixture—can be significantly improved. With the further research on factors not considered in this study, such as the molecular weight, supercritical regime, and effects of a minus saturation temperature, it is expected that a comprehensive model for the PMMA and ${\mathrm{CO}}_{2}$ mixture can be proposed.

## Figures and Tables

**Figure 1 polymers-14-00596-f001:**
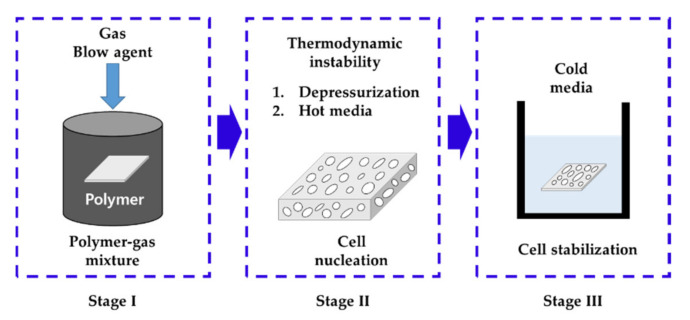
Theoretical principle for the solid-state batch foaming process.

**Figure 2 polymers-14-00596-f002:**
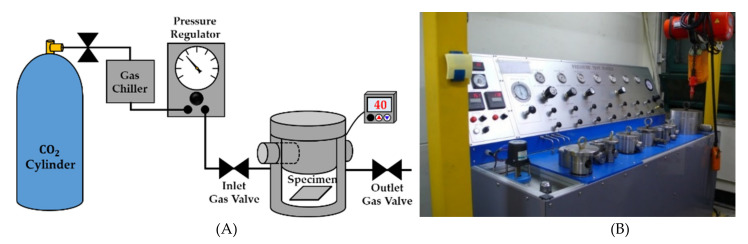
(**A**) Schematic of the solid-state batch foaming process; (**B**) experimental equipment of the solid-state batch foaming process used in this study.

**Figure 3 polymers-14-00596-f003:**
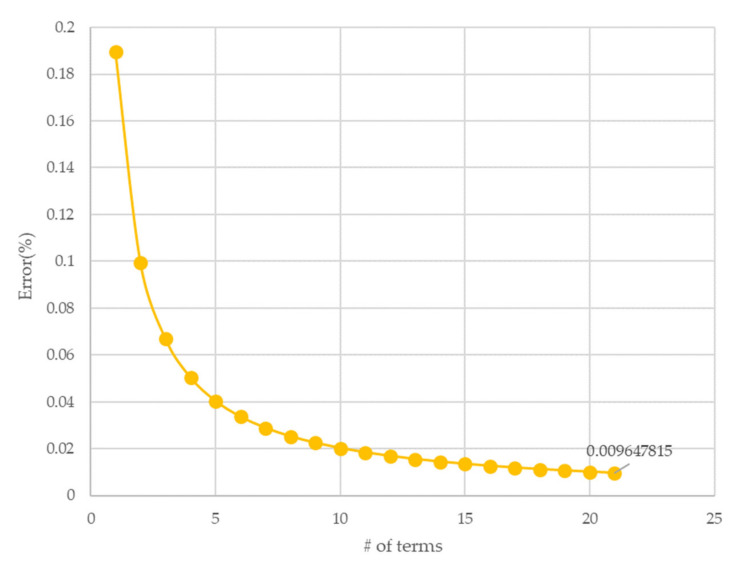
Number of terms versus error in Equation (2) at the initial time.

**Figure 4 polymers-14-00596-f004:**
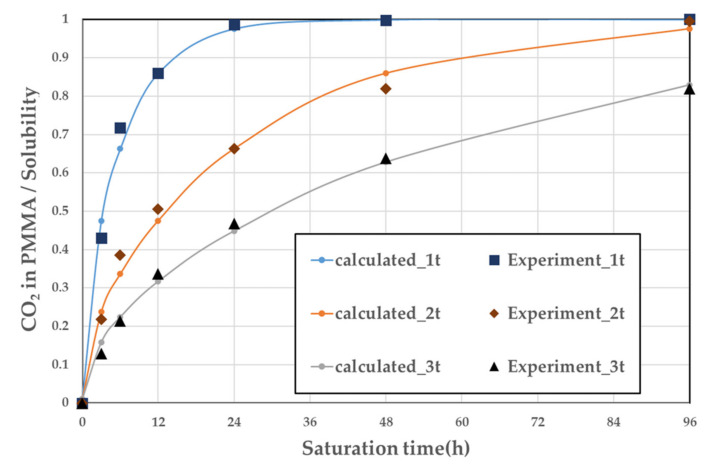
Sorption curves at 5 MPa and a temperature of 290 K.

**Figure 5 polymers-14-00596-f005:**
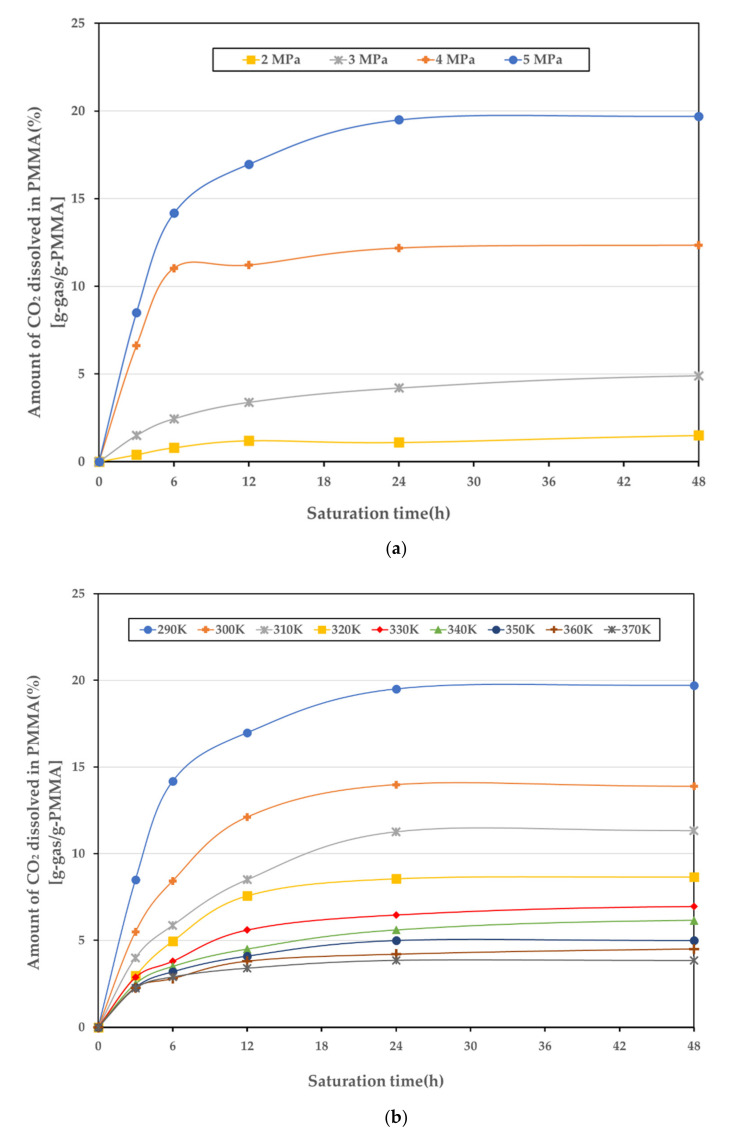
Sorption curves; (**a**) 5 MPa for temperatures in the range 290–370 K; (**b**) 290 K for pressures in the range 2–5 MPa.

**Figure 6 polymers-14-00596-f006:**
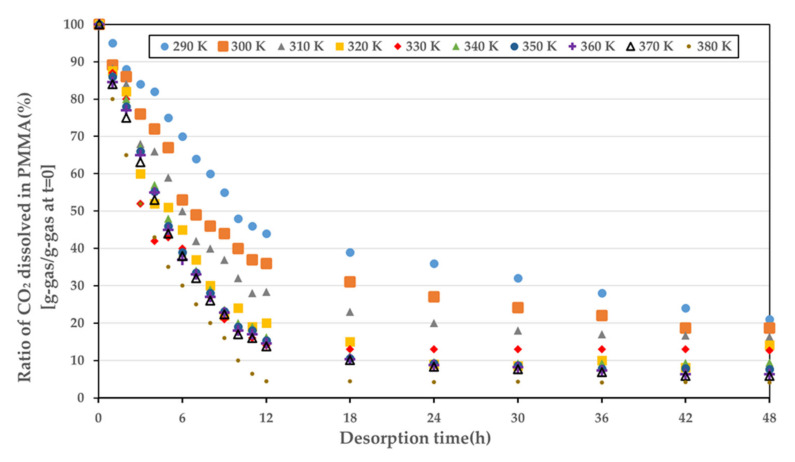
Desorption curves for various temperatures at atmospheric pressure.

**Figure 7 polymers-14-00596-f007:**
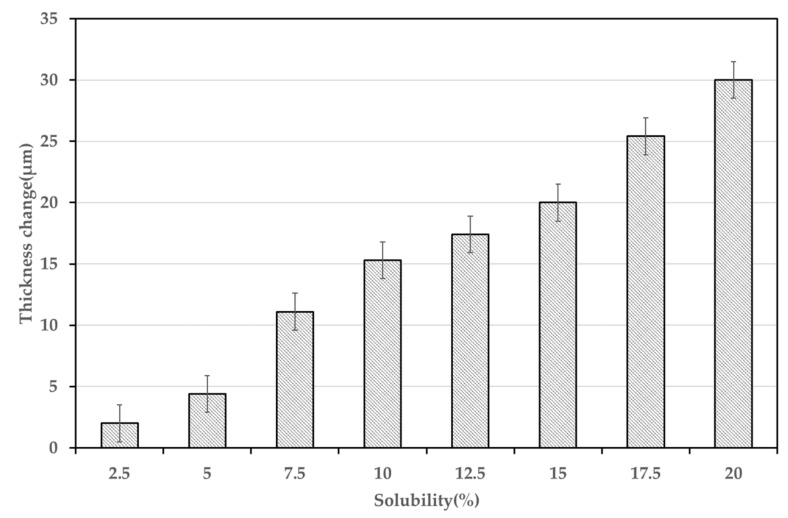
Average swelling depending on the solubility.

**Figure 8 polymers-14-00596-f008:**
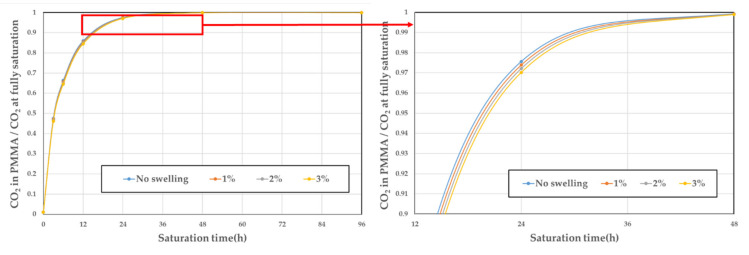
Sorption curves relative to the change in thickness using the same diffusion coefficient.

**Figure 9 polymers-14-00596-f009:**
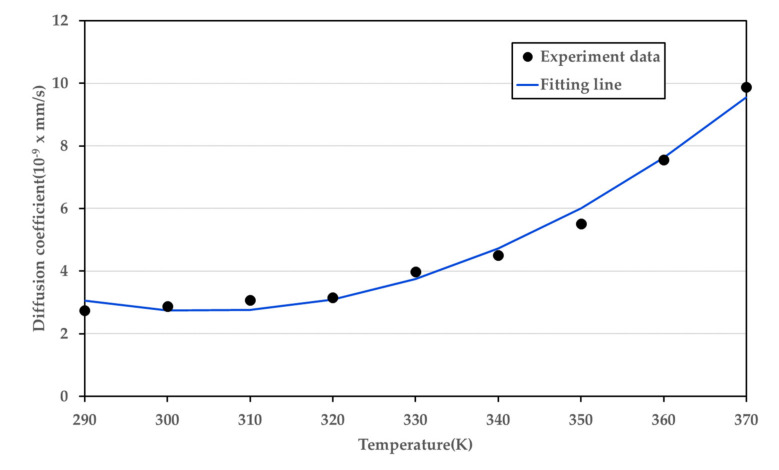
Diffusion coefficient of the sorption process at various temperatures and pressures.

**Figure 10 polymers-14-00596-f010:**
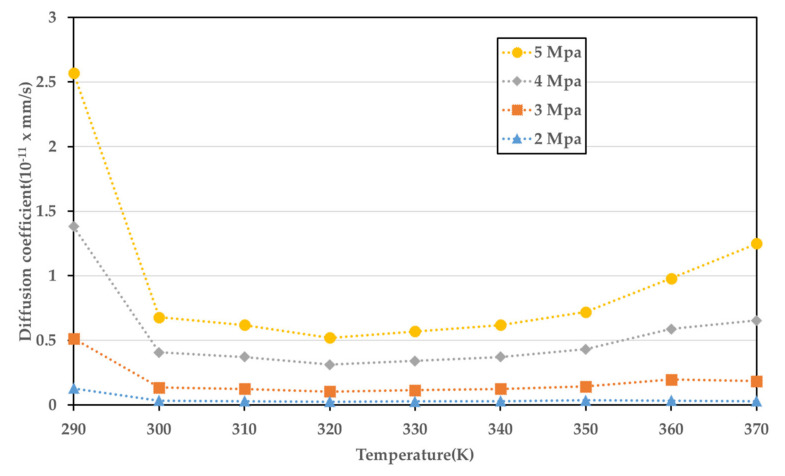
Diffusion coefficient of the desorption process at various temperatures.

**Figure 11 polymers-14-00596-f011:**
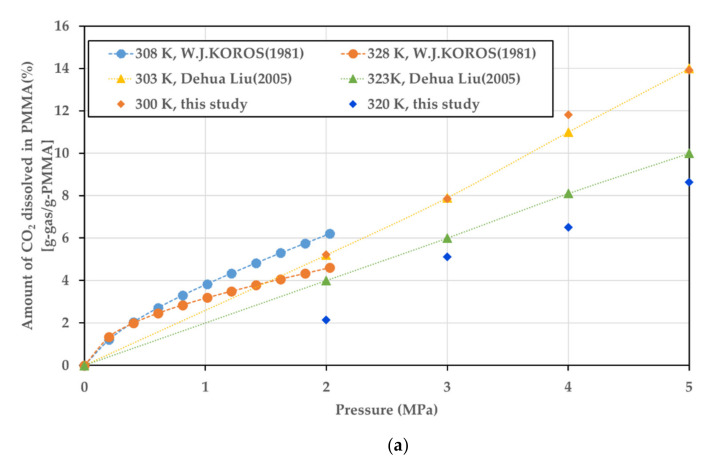
(**a**) Comparison of CO_2_ sorption in PMMA in this study and the studies of Koros et al. [32] and Liu et al. [33]. (**b**) Comparison of diffusion coefficient of the PMMA–CO_2_ mixture in this study and that in the study of Guo et al. [34].

**Table 1 polymers-14-00596-t001:** Diffusion coefficient at 5 MPa and a temperature of 290 K.

	1 mm	2 mm	3 mm
$\mathrm{Diffusion}\;\mathrm{coefficient}\;(10^{-11}\times {\mathrm{mm}}^{2}$/s)	2.31	2.25	2.28
RMS error (%)	0.053	0.187	0.084

**Table 2 polymers-14-00596-t002:** Solubility at various saturation temperatures and pressures.

	P_sat_	2 MPa	3 MPa	4 MPa	5 MPa
T_sat_					
290 K		2.05%	5.18%	12.4%	19.8%
300 K		1.68%	3.71%	9.24%	13.9%
310 K		1.54%	2.9%	6.54%	11.3%
320 K		1.24%	2.21%	5.31%	8.63%
330 K		0.99%	1.85%	3.87%	6.95%
340 K		0.95%	1.6%	3.15%	6.14%
350 K		0.91%	1.32%	2.97%	5.04%
360 K		0.85%	1.24%	2.91%	4.81%
370 K		0.71%	1.09%	2.84%	3.85%

## Data Availability

Not applicable.

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
