# Peer review of "Modeling and Experiment for the Diffusion Coefficient of Subcritical Carbon Dioxide in Poly(methyl methacrylate) to Predict Gas Sorption and Desorption"

_polymers, 2022, doi:10.3390/polym14030596_

Round 1

Reviewer 1 Report

In this manuscript, the authors studied the gas sorption and desorption experiments of poly(methyl methacrylate) (PMMA) using solid-state batch foaming and the weight change as a function of temperature and pressure. The diffusion coefficient of the PMMA was calculated using Fick’s law and also the diffusion coefficient model was employed using the least-squares method. Subsequently, the model was validated by comparing it to the literature values, and the overall trend was found to be consistent. Therefore, it was confirmed that there was little difference between the diffusion coefficient obtained using only Fick's equation and the value obtained by a different method. This work is very good and useful for the field of microcellular foaming process (MCPs) technologies that can be used throughout the industries, including automobiles, sports, and packaging, as it has the advantages of specific strength, thermal insulation, reflectance, acoustic absorption, electromagnetic interference shield, and lightweight ordinary plastic foaming. The interpretations and conclusions are well justified by the results. In addition, the quality and quantity of the figures are appropriate. The references are well related to work and up to date.

Summary: I recommend publishing this manuscript after taking into account my comments on the attached file.

Reviewer 2 Report

Reviewer Report

In this manuscript, the authors report on the gas sorption and desorption experiments of poly(methyl methacrylate) (PMMA) which were conducted via solid-state batch foaming and the weight change was measured.  The control parameters were temperature and pressure.

Based on the experimental data, the diffusion coefficient of the PMMA was calculated using Fick’s law. Since the manuscript reports interesting and important results in the field, my recommendation is to accept it for publication in Polymers Journal, subject to the following minor revision point:

  • Equation (1) is not diffusion equation, as authors claim, it is solution of diffusion equation. That diffusion equation should be mentioned and discussed before presenting its solution (1).
  • Authors should give more comments on steady-state conditions for CO2 concentrations for different values of the parameters (such as pressure).
  • It would be interesting if authors could mention some relevant works which investigated the characteristics of PMMA under external influence (temperature, gamma radiation), such as:

Al-Kadhemy, M.F.H., Saeed, A.A., Khaleel, R.I. et al. Effect of gamma ray on optical characteristics of (PMMA/PS) polymer blends. J Theor Appl Phys 11, 201–207 (2017).

M. S. Kovacevic, A.  Djordjevich, Savovic, J.  S.  Bajic, D.  Z.  Stupar, M.  P. Slankamenac,   M.   Kovacevic,   Measurement   of   60Co   gamma   radiation   induced attenuation in multimode step-index POF at 530 nm, Nuclear Technology and Radiation Protection, Vol. 28,  No. 2,  2013, pp. 158-162.

S. Savovic, M. S. Kovacevic, J. S. Bajic, D. Z. Stupar, A. Djordjevich, M. Zivanov, B. Drljaca, A. Simovic, K. Oh, Temperature dependence of mode coupling in low-NA plastic optical fibers,   Journal  of  Lightwave  Technology,  Vol.  33,   No. 1,  2015,  pp. 89- 94.
